# "Testing for malaria does not cure any pain" A qualitative study exploring low use of malaria rapid diagnostic tests at drug shops in rural Uganda

**Victoria Shelus**[1,2]*, **Nobert Mumbere**[3], **Amos Masereka**[3], **Bonita Masika**[3], **Joackim Kiitha**[3], **Grace Nyangoma**[3], **Edgar M. Mulogo**[3], **Clare Barrington**[1,2], **Emmanuel Baguma**[3], **Rabbison Muhindo**[3], **James E. Herrington, Jr.**[1], **Michael Emch**[2,4,5], **Suzanne Maman**[1], **Ross M. Boyce**[2,4,6]

**1** Department of Health Behavior, Gillings School of Global Public Health, University of North Carolina at Chapel Hill, Chapel Hill, North Carolina, United States of America, **2** Carolina Population Center, Chapel Hill, North Carolina, United States of America, **3** Department of Community Health, Faculty of Medicine, Mbarara University of Science and Technology, Mbarara, Uganda, **4** Department of Epidemiology, Gillings School of Global Public Health, University of North Carolina at Chapel Hill, Chapel Hill, North Carolina, United States of America, **5** Department of Geography, University of North Carolina at Chapel Hill, Chapel Hill, North Carolina, United States of America, **6** Institute for Global Health and Infectious Diseases, University of North Carolina at Chapel Hill, Chapel Hill, North Carolina, United States of America

* Victoria.Shelus@gmail.com

**Data Availability Statement:** The data presented in this study are not publicly available because public

## Abstract

The World Health Organization recommends all suspected malaria cases be confirmed with a parasitological test, typically a rapid diagnostic test (RDT), prior to treatment. Despite recommendations, many fevers presenting at private drug shops are treated presumptively as malaria without diagnostic testing. The purpose of this qualitative study was to describe community perceptions of RDTs and explore ways to improve malaria case management at drug shops in Bugoye, western Uganda. A total of 63 in-depth interviews were conducted between September and December 2021 with 24 drug shop clients, 19 drug shop vendors, 12 community health workers, and 8 health and community officials. Data was analyzed using thematic content analysis and narrative techniques. While drug shop clients valued RDTs, the cost of the test limited their use. Further, mistrust in negative results and fear about treatment options for conditions other than malaria led to nonadherence to negative RDTs. Improvement with antimalarials after a negative RDT, or no RDT at all, was seen as proof an individual had malaria, reinforcing the acceptability of liberal antimalarial use. Drug shop vendors were knowledgeable about malaria case management but financially conflicted between recommending best practices and losing business. While clients viewed drug shop vendors as trusted health professionals, health officials distrusted them as business owners focused on maximizing profits. Study results suggest public-private partnerships that recognize the essential role of drug shops, better incorporate them into the healthcare system, and leverage the high levels of community trust in vendors, could provide greater opportunities for oversight and training to improve private-sector malaria case management. Interventions that address financial barriers to RDT use, emphasize the

deposition would breach compliance with the protocol approved by the research ethics board. The dataset was made available to facilitate peer review. Researchers may apply for data access by contacting the corresponding author or the UNC IRB at irb_questions@unc.edu.

**Funding:** This work was supported by the U.S. Department of State, Bureau of Educational and Cultural Affairs [Fulbright-Fogarty Award in Public Health E0636820 to V.S], the Doris Duke Charitable Foundation [Caregivers at Carolina Award 2015213 to R.M.B.], and the National Institutes of Health [K23AI141764 to R.M.B.]. The funders had no role in study design, data collection and analysis, decision to publish, or preparation of the manuscript.

**Competing interests:** The authors have declared that no competing interests exist.

financial benefits of malaria testing, increase vendor knowledge about illnesses confused with malaria, and improve the quality of vendor-client counseling could increase RDT uptake and improve adherence to RDT results.

## Introduction

Malaria is a persistent public health threat to nearly half of the world's population [1]. Globally, there were an estimated 241 million malaria cases and 627,000 deaths attributed to malaria in 2020, the majority in children under 5 years of age in sub-Saharan Africa [2]. Uganda is ranked third in the world for total number of malaria cases [2], where *Plasmodium falciparum* malaria is the leading cause of morbidity and mortality among all ages [3]. While malaria prevalence had been steadily declining as a result of preventive measures, such as insecticide-treated bed nets, annual malaria cases in Uganda increased from 11.6 to 13.0 million between 2019 and 2020 [2], likely due to COVID-19-related disruptions in malaria prevention and treatment services [4]. The high prevalence of malaria in Uganda places a substantial burden on the health system [1], and has a negative socio-economic impact, with households spending a considerable proportion of their income on malaria prevention and treatment [3].

Private drug shops in Uganda are commonly the first point of care for fever and an important source of essential medications such as antimalarials [5, 6]. Despite their popularity, many drug shops are unlicensed or operated by vendors who lack the qualifications required by the National Drug Authority [7–9]. Drug shop vendors do not always have formal training in malaria case management [10] or provide treatment in accordance with national guidelines [5, 7, 9–12]. Perhaps most critically, not all clients receiving antimalarials from drug shops have a confirmed malaria diagnosis. While vendors rely on presumptive treatment of fevers as malaria [13], the nonspecific symptoms of malaria, such as fever, headache, and myalgia, make clinical diagnosis difficult [14]. Presumptive treatment contributes to overtreatment for malaria, which drives parasite resistance, wastes limited resources, and could lead to delays in the diagnosis and treatment for other potentially serious infections [14–17].

Malaria rapid diagnostic test (RDT) availability in drug shops offers a tool for the timely and accurate diagnosis of malaria, by detecting the presence of specific malaria antigens from a capillary (i.e., finger-prick) blood sample. RDTs quickly diagnose malaria in settings where high-quality laboratory services are not routinely available, and can reduce antimalarial overprescriptions [18, 19]. Both the World Health Organization and the Ministry of Health of Uganda promote "test and treat" guidelines, which recommend all suspected malaria cases be confirmed with a diagnostic test prior to treatment [3, 20]. While RDT use in public facilities in Uganda is high, with an estimated diagnostic testing coverage of 84% [1], RDTs remain underused in the private sector. Previous studies found only 10 to 30% of clients treated for malaria at drug shops received an RDT [11–13]. Even when RDTs are used at the drug shop, clients often purchase antimalarials despite a negative result [13].

Several programs in Uganda have increased the use of RDTs at drug shops, and improved the quality of malaria case management more generally, through vendor trainings, community-level demand generation activities, provision of free or subsidized RDTs, and peer supervision [21–26]. While most studies were pilot interventions focused on RDT introduction [21–24], these diagnostic tests are now commonly stocked at drug shops [27]. More research on strategies to improve testing of suspected malaria cases prior to treatment is needed in a context where RDTs are readily available but infrequently used.

Understanding the root causes of low RDT use and subsequent malaria overtreatment is needed to develop effective responses. Qualitative research in Uganda suggests that while

community members are generally supportive of RDTs, use is limited by financial barriers, low perceived need for diagnosis, and negative perceptions about the credibility of RDTs and vendors [28–30]. Perspectives of key stakeholders, such as vendors, health officials, and health workers, are also needed to ensure recommendations are feasible and comprehensive. For example, while officials may be concerned about intervening with unlicensed drug shops, vendors have a vested interest in their own financial security. The early involvement of stakeholders may increase their buy-in to future interventions, helping to ensure program success and sustainability [31, 32].

The purpose of this cross-sectional qualitative study was to describe community perceptions of RDTs and explore ways to improve malaria case management at drug shops in Bugoye, western Uganda. In-depth interviews were conducted with drug shop clients, drug shop vendors, community health workers, and health and community officials. This paper provides intervention recommendations based on an understanding of the root causes of overtreatment for malaria in this setting, and the diverse perceptions of key stakeholders.

## Methods

### Study setting

Bugoye is a rural sub-county located in Kasese district in western Uganda with a population of approximately 42,000 residents. More than 80% of households rely on subsistence farming for their livelihood [33]. The tropical climate allows for year-round malaria transmission interspersed with semi-annual transmission peaks after the rainy seasons (March to May and September to November) [34]. In the public sector, free health care services are available from six level II health centers staffed by nurses and midwives, and two level III health centers staffed by clinical officers. However, approximately 22% of households are 5 km or further from the nearest health center, often through steep, hillside terrain [33]. In villages, community health workers, known as Village Health Team members (VHTs), treat uncomplicated pneumonia, diarrhea, and malaria in children under five. Given the remote nature of many of the communities in Bugoye, private drug shops play a large role in the distribution of antimalarials [27].

### Malaria case management at drug shops in Bugoye

Forty-six drug shops were operating in Bugoye in 2021, and all participated in a related quantitative study at the same site which documented malaria diagnostic and treatment practices, and informed the sampling strategy and qualitative themes for the current study [35]. Most drug shops were unlicensed (96%), and many vendors were trained as nursing assistants (70%), which does not meet the necessary qualifications to operate a drug shop in Uganda. Nearly all drug shops sold RDTs (98%) at a median price of 2000 Ugandan Shillings (approximately $0.57). Despite widespread availability, there was low use of RDTs at drug shops. Most clients (60%) were not aware of their malaria status at the drug shop while purchasing medications. Additionally, there was low adherence to negative RDT results. More than one-third of clients who tested negative still purchased antimalarials (36%). Many clients who purchased an antimalarial without an RDT subsequently tested negative for malaria (65%). Family members often sought treatment for someone who was sick at home and therefore unable to be tested.

### Sample & recruitment

Individuals were eligible to participate in the study if they were 18 years or older, spoke the local language of Ihukonzo or English, and met one of the following criteria: (1) client of a drug shop in Bugoye with suspected malaria in the two weeks prior; (2) own/work at drug

shop in Bugoye for at least 2 years; (3) VHT in Bugoye for at least 2 years; or (4) health or community official with a professional stake in interventions to improve malaria case management at drug shops. Target sample sizes were based on qualitative methods research that found saturation occurs between six and twelve interviews within a homogenous sample [36]. We aimed to interview six individuals from each stratum within participant categories.

Drug shop clients were recruited from a purposive sample of drug shops to achieve geographic diversity between remote villages and trading centers, defined by the presence of a weekly market day. Vendors received a paper form to record the names and contact information of clients who purchased an antimalarial from their shop and were willing to be contacted by the study team for an interview. Clients were stratified based on sex and whether an RDT was purchased at the drug shop. Drug shop vendors were recruited to represent four groups with varying levels of RDT use: vendors at shops that did not sell RDTs, vendors at shops with low RDT sales (less than 20% of clients with suspected malaria bought an RDT), vendors at shops with medium RDT sales (20–40% of clients), and vendors at shops with high RDT sales (greater than 40% of clients). Vendors were purposively selected to achieve geographic variation within the sample. VHTs were randomly selected from a list of all 174 VHTs in Bugoye and stratified by sex. Health and community officials were individually selected based on study team discussions about individuals at the sub-county and district level with a stake in the design of future interventions to improve malaria case management at drug shops in Bugoye. The study team called identified individuals to briefly describe the study, confirm interest and eligibility, and schedule an interview.

## Data collection

A total of 63 interviews were conducted between September and December 2021 with 24 drug shop clients, 19 drug shop vendors, 12 VHTs, and 8 health and community officials (Table 1). Clients and vendors were evenly sampled from drug shops in one of the three major trading centers in the sub-county, and drug shops from more remote villages. Interviews were

**Table 1. Description of study participants (n = 63).**

| Participant category | Category sample size | Strata | n |
|---|---|---|---|
| Drug shop clients | 24 | Men who purchased an RDT | 7 |
| | | Men who did not purchase an RDT | 6 |
| | | Women who purchased an RDT | 6 |
| | | Women who did not purchase an RDT | 5 |
| Drug shop vendors* | 19 | Shop with no RDT sales | 1 |
| | | Shop with low RDT sales | 6 |
| | | Shop with medium RDT sales | 6 |
| | | Shop with high RDT sales | 6 |
| Village Health Team members | 12 | Men | 6 |
| | | Women | 6 |
| Health and community officials | 8 | Sub-county level | 3 |
| | | District-level | 5 |

*Previous study results [35] were used to determine cut-points for RDT sales. There was only one drug shop in the sub-county that did not sell RDTs. Drug shops were defined as having low RDT sales if less than 20% of clients with suspected malaria) bought an RDT. At drug shops with medium RDT sales, 20–40% of clients with suspected malaria bought an RDT. At drug shops with high RDT sales, greater than 40% of clients with suspected malaria bought an RDT.

conducted in Ihukonzo or English. Local interviewers were from the region, had experience conducting research with low-literate participants, and received training on study procedures, qualitative research best practices, and research ethics. All interviews took place at a convenient time and private location of the participant's choice, lasted approximately one hour, and were recorded and transcribed in English.

Interviews were conducted using semi-structured in-depth interview guides that included open-ended questions and suggested probes (S1 Appendix); however, interviewers had the flexibility to explore themes that arose naturally during the conversation. Guides were informed by theoretical constructs from the Behavioral Model of Health Services Use [37] and Diffusion of Innovations [38, 39], and factors known to influence client and vendor perspectives on RDTs from the literature.

Interviews with all participant categories explored perceptions of malaria, drug shops, and RDTs, and interviews with vendors, VHTs, and officials asked about suggestions for interventions (Table 2). A one-page summary of drug shop practices in Bugoye was shared with officials during the interview to gauge their reactions and guide the discussion (S2 Appendix). Interactions between clients and vendors at the drug shop was a focus of interviews with both clients and vendors. Client interviews also explored decision-making around RDT and medication purchases at the drug shop; previous experiences with diagnosis and treatment of malaria; and trusted sources of health information. Interviews with drug shop vendors included history of RDT use at the drug shop, and training and sources of information on RDTs.

## Analysis

Qualitative thematic content analysis techniques were used to analyze the data, following an iterative process of reading, coding, data display, and reduction. Debrief forms were completed

**Table 2. Themes covered in qualitative interviews, by participant category.**

| Participant category | Themes |
|---|---|
| Drug shop clients | Perceptions of malaria and RDTs |
| | Illness history |
| | Decision to go to the drug shop |
| | Interactions with the drug shop vendor |
| | RDT and medication purchases at the drug shop |
| | Previous experiences with diagnosis and treatment of malaria |
| | Trusted sources of health information |
| Drug shop vendors | History of RDT use at the drug shop |
| | Training and information on RDTs |
| | Perceptions of RDTs |
| | Interactions with drug shop clients |
| | Use of RDTs at drug shops |
| | Adherence to RDT results at drug shops |
| | Perceptions of proposed intervention options |
| Village Health Team members | Perceptions of malaria |
| | Perceptions of RDTs |
| | Interactions with patients |
| | Perceptions of proposed intervention options |
| Health and community officials | Health challenges in the community |
| | Current regulation of drug shops |
| | Review and reactions to drug shop practices |
| | Perceptions of proposed intervention options |

after each interview to capture key themes (S3 Appendix). Review of debrief forms and interview transcripts and team discussions during fieldwork were used to refine in-depth interview guides to ensure that a range of perspectives were being captured and to confirm saturation had been reached within each category of interview participants. A codebook was developed for retrieving text related to interview questions and emerging themes (S4 Appendix). MAXQDA 2020 (VERBI Software) was used to code interview transcripts. Primary coding reports were extracted and further analyzed. Memos were developed to summarize findings within each broad theme and compare groups within each category of participants. To supplement thematic analysis, narrative summaries at the participant-level were written by extracting key storylines from client transcripts to describe individual experiences with malaria illness and seeking care [40]. References to verbatim text support interpretation of thematic relationships and provide additional explanation to patterns observed. Illustrative quotes included in this paper contain sex, participant category, and stratum for context.

### Ethical considerations

This study was approved by the University of North Carolina Office of Human Research Ethics (20–3019), the Mbarara University of Science and Technology Research Ethics Committee (MUST-2021-55), and the Uganda National Council for Science and Technology. All participants provided written informed consent. As a token of appreciation for their participation in the study, health and community officials received a small monetary incentive of $7 (subcounty level) or $11 (district-level). Clients, vendors, and VHTs received soap and sugar, valued at approximately $1.35.

## Results

We first characterize participants' perceptions of the role of drug shops in their communities. Next, we describe perceptions of RDTs, specifically whether they are viewed as a necessary component of malaria care and the results are trusted. Then, we explore RDT use, adherence to test results, and reactions to practices not in accordance with national guidelines. Finally, we discuss participants' recommendations for the best ways to intervene and improve malaria case management at drug shops.

### Role of drug shops

Most drug shop clients came to the drug shop as their initial point of care for fever, while four had previously been to a health center and then sought additional medications or testing at a drug shop. Clients frequently used drug shops rather than a public health facility because they needed treatment when public health centers were closed. If symptoms began or worsened in the evening or over the weekend, even clients who preferred health centers bought medication from a drug shop because it was the only outpatient option available at those times for individuals over the age of five. Other clients expressed a preference for drug shops. They described how government health centers often have long wait times and medication stockouts, are located far away and require money for transport, and are unable to provide the same level of personalized customer service as drug shops.

> *At the public facility, there is always a long line. You sweat and suffer, and after you have been diagnosed, the nurses will tell you they don't have the medicines to give you. When you have some money, it is better you buy medicines from the drug shop.* (Female client, no RDT)

The frustration this client expressed was a common sentiment. Many viewed it as a waste of time to go to the health center if you would end up being referred to a drug shop to purchase out-of-stock medications.

Given the number of drug shops, clients have a choice about where to seek treatment, and expressed strong preferences for particular shops and vendors. Proximity of the drug shop and vendor characteristics, such as being friendly and polite, were important considerations when selecting a drug shop. Clients viewed vendors as trusted and educated members of the community.

> *I have belief in that drug shop attendant because whenever I buy medication from him, I get cured, and he is a good friend of mine. There are other drug shops around, but I feel like they cannot treat my illnesses.* (Male client, no RDT)

In addition to vendors' characteristics, as demonstrated in this quote, many clients always went to the same shop because they had a history of receiving medications there and recovering, and as a result they developed a high level of confidence in the vendor.

In contrast to client perceptions, health officials and VHTs raised many concerns about drug shops and vendors. Most officials and a few VHTs recognized drug shop vendors primarily as business owners rather than health professionals, and they attributed poor practices at drug shops to the goal of private businesses to maximize profits. "*These are traders. Traders are more inclined on profits. They're not bothered whether they're doing it the right way or the wrong way*" (Health official, sub-county level). Despite these concerns, health officials and VHTs generally recognized the important role drug shops play in supplying essential medications, treating illnesses at the community-level, and addressing health challenges such as malaria, but emphasized that vendors must be qualified.

### Perceptions of rapid diagnostic tests (RDTs)

Drug shop vendors and VHTs had overwhelmingly favorable opinions of RDTs. As expressed in the quote below, they appreciated that RDTs eliminate the guesswork associated with symptomatic malaria diagnoses and allow them to distinguish between malaria and other causes of fever.

> *When you use an RDT and it reads positive, you are capable of treating the patient and they quickly respond. If it reads negative, you see what other ways to treat the patient depending on the symptoms. . .When you diagnose clinically, you will always have doubt about everything. You even start wondering if it's typhoid or malaria or something else.* (Male vendor, high RDT use)

By quickly providing information to direct treatment decisions, RDTs allow vendors to treat malaria with confidence so clients recover faster. Vendors reported RDTs were easy to use with only minor challenges–such as collecting blood from fearful children or having to repeat RDTs with invalid results. A few vendors indicated that stocking RDTs was also financially advantageous as they profited from RDT sales. One vendor believed RDTs had garnered him respect, which ultimately improved his business, "*A client simply says, 'I am going to that man's drug shop, he always first tests you before giving you medication.' It has helped me to get big market" (*Male vendor, low RDT use).

Drug shop clients also had positive opinions about RDTs, and most thought RDTs were necessary to identify the cause of symptoms. Only two clients felt that RDTs were not needed, believing you can tell whether you have malaria based on symptoms alone. As reflected in the

quotes below, clients expressed positive views regardless of whether they had recently purchased an RDT at the drug shop.

> *I feel it is good to use an RDT because if you are sick, you will know that you have malaria, so you will be able to buy drugs for malaria. And this will help you to recover in time, because you are taking the medicine for the disease [you have]*. (Female client, RDT)

> *These [RDTs] help us to know what we are suffering from. If we don't have them, then we can't be able to know how sick we are, and what we are suffering from*. (Male client, no RDT)

Despite mostly positive perceptions of RDTs, vendors and clients did not always have full confidence in results, especially if they were negative. The primary complaint about RDTs among both vendors and clients was the expectation of false negatives if the disease was in its early stages, if someone had taken antimalarials prior to testing, or if the malaria type was something other than *P. falciparum*. These beliefs contributed to concerns about RDT accuracy more generally, as reflected below.

> *Sometimes the RDT fails to detect malaria even when a person is sick. However, on the other hand, an RDT is a good test because it does show true results in most times*. (Male client, RDT)

> *RDTs do not tell lies. If it tells you negative then it is negative, if it says positive then it is positive. But now again sometimes a person may come, and they describe to you the signs and symptoms of malaria, but when you use an RDT, it shows you that the person is negative. And so there I keep asking myself what could have happened*? (Female vendor, low RDT use)

These quotes show how even vendors and clients who said they believe RDTs are accurate sometimes question their accuracy when someone with all the signs and symptoms of malaria tests negative.

## Use of RDTs

Among the thirteen clients who purchased RDTs, four made the decision to be tested on their own, before coming to the drug shop. However, most agreed to be tested based on the vendor's recommendation. These clients purchased an RDT after the vendor suggested it, indicating the importance of client-vendor discussion about malaria testing.

> *The drug shop vendor did suggest that I get tested because the signs I had explained to him showed that I could be sick of malaria. . .I agreed because I was feeling sickly, and I wanted to know what I was really sick of*. (Male client, RDT).

This quote also highlights the two most common reasons clients bought an RDT at the drug shop: they felt sick, with symptoms severe enough to cause concern, and they wanted to know whether their symptoms were caused by malaria. A few clients wanted an RDT to take the right medication for their illness or confirm a previous negative RDT result, or they were encouraged by vendor flexibility with the timing of payment. One client, who made the decision to get tested prior to arriving at the drug shop, got an RDT because of a previous experience where typhoid was misdiagnosed as malaria.

For clients who purchased antimalarials at the drug shop without an RDT, the most common reason for forgoing diagnostic testing was the cost of the test.

*I did not have enough money to carry out a malaria test. That day I had 2000 shillings, it was not enough even to purchase my medication. . .Testing for malaria does not cure any pain, so I would rather reduce the pain I am feeling than do a malaria test.* (Male client, no RDT)

As reflected in this quote, clients often came to the drug shop with only enough money for medicine, if that. With limited resources, clients prioritized treatment over testing: "*When I have no money, I definitely don't get an RDT, I'd rather buy medicine only*" (Male client, no RDT). A few clients did not get tested because they were buying medications on behalf of someone else who was sick at home, or because they didn't see the value of an RDT because they had the signs and symptoms of malaria. Many of these clients, however, indicated they had bought RDTs in the past, suggesting that even if RDTs are valued, financial constraints limit their use at drug shops.

Drug shop vendors provided additional explanations for the low use of RDTs. First, clients who came with results from a health center didn't need to buy RDTs. Additionally, some clients express fear of the finger prick or that the RDT tests for other illnesses (e.g., HIV), though these were infrequent. Nearly every vendor echoed that cost was the main barrier to buying RDTs: "*The community has already known the importance of RDTs. . .The truth is they do not have the money*" (Female vendor, high RDT use). A few vendors said RDT use varies depending on the agricultural season, which has implications for the amount of disposable income that clients have available for drug shop purchases.

*If people are well-off like in this coffee season, patients themselves come and request you to test them for malaria if they are not feeling well. . .But if it's a season where people don't have money, it's hard to find people testing for malaria.* (Female vendor, medium RDT use)

## Adherence to RDT results

While none of the interviewed clients purchased antimalarials after receiving a negative RDT during their most recent drug shop visit, two described past experiences when they did not adhere to their test results.

*When she tested me, I turned out to be negative. But when she looked at how I was looking sick and how I was complaining about feeling so bad, she just gave me Coartem [antimalarial]. . .I took the Coartem the nurse had given me, and I got better. That means I was positive with malaria though the RDT showed I was negative.* (Female client, no RDT)

This quote highlights that the use of antimalarials after a negative RDT stems from distrust in negative results, from both vendors and clients. Additionally, it exemplifies the pervasive theme that improvement with antimalarials after a negative RDT, or no RDT at all, was "proof" that an individual had malaria. This belief reinforces the acceptability of these practices, even though individuals may recover on their own regardless of treatment, or because of other medications taken, such as antibiotics or painkillers.

Vendors reported that clients bought antimalarials after testing negative because they still believed they had malaria based on their symptoms, they didn't understand what the medication treats, or out of panic and desperation to try anything to feel better. Vendors explained that when RDTs are positive, clients are relieved to know the cause of their symptoms and hopeful they will heal with medication. Because clients are expecting a positive result based on their symptoms, it is not difficult to accept the result as truth. However, when RDT results are negative, clients are scared and uncertain about the next steps.

*I felt happy because when I tested positive for malaria. . .I was given the medicine. I then went home, took the drugs, and I felt fine after 2 to 3 days. . .[Another time], I was tested negative for malaria. I asked them, 'What is wrong with me?'. . .I felt worried because I was sick and I was expecting to have malaria, but they told me I was negative for malaria.* (Male client, no RDT)

This client's fear and confusion after testing negative for malaria was a common experience. In response, vendors reported trying to explain that there are many causes of fever, and to identify and treat the cause of the fever, but most believed that some or all their clients still did not accept negative test results.

Although vendors recognized that the use of antimalarials without an RDT or after a negative test result are problematic, only a few insisted on the use of RDTs, referred clients to the health center, or attempted to explain the importance of using RDTs or only taking antimalarials if you test positive. If clients requested antimalarials without an RDT or after a negative result, most vendors simply provided it.

*If you say, 'Let me use my profession. If you are negative, I will not give you medicine.' That client will go and tell others that you don't give clients treatment. . .What you do, you simply treat them as they wish, and they take a good report about you. But if you decide to use professionalism, you get challenge of losing customers.* (Male vendor, low RDT use)

This quote demonstrates the tension between drug shop vendors as health providers and private business owners. Vendors felt torn between recommending the best practices for clients and losing business. While some clients purchased medications based on the vendor's recommendation, others asked for specific medications. Vendors were reluctant to lose money from the sale of medications requested by clients, even if they were not recommended based on national treatment guidelines.

## Intervention recommendations

Health officials, drug shop vendors, and VHTs discussed five recommendations to improve malaria case management at drug shops: 1) community sensitization, 2) vendor trainings, 3) provision of free or subsidized RDTs, 4) increased enforcement, and 5) better integration of drug shops into the public healthcare sector.

Community education was the most frequent intervention suggested. Participants explained that there is an existing practice of trusted local leaders and VHTs sharing messages at community events. Participants believed sensitization focused specifically on the importance of RDTs could address a knowledge gap among clients, and drug shop vendors were supportive of any education efforts that could increase RDT sales. However, some participants recognized the inherent difficulties associated with changing behavior, and that sensitization alone would not address financial barriers to RDT use.

Participants suggested sharing messages with the community such as: it's important to test to know the cause of your illness, taking medications you don't need can cause problems, and the person who is sick should go to the drug shop so they can be tested. A few vendors and VHTs also believed messages about how RDTs can save you time and money would resonate. While some health officials and VHTs suggested messages encouraging community members to seek treatment from public health centers rather than drug shops, vendors cautioned that such an approach could have unintended consequences.

*You may hear people saying, 'People should go and get tested. Don't go to fake or unqualified health workers.' And for us here in the community, we have helped so many people. As they*

*teach like that, the person that sees how much drug shops help people in the community here, the person does not believe in what is being taught. . .It's better that as they sensitize, they tell people that every facility you visit, make sure you get an RDT.* (Male vendor, low RDT use)

As this vendor suggested, sensitization efforts should focus on the importance of testing no matter where you seek treatment, rather than diminishing drug shops.

Training for drug shop vendors on general malaria diagnosis and treatment, new treatment guidelines, and when to refer clients to health facilities was another suggested intervention. Health officials and VHTs viewed vendor trainings as a necessary complement to community sensitization. Trainings were acceptable to vendors because they would benefit by gaining new knowledge and standardizing practices across shops based on the latest government guidelines, which would lead to better services. There was also a recognized need among vendors and health officials for additional training and practice on how to counsel drug shop clients.

*Many drug shop vendors lack counseling skills. . .Testing is supposed to be done in the laboratory where we have a lab technician and a counselor who discloses the results to the clients. This counselor is highly trained to do that job, but for us we are not. I may test someone and find they are negative, and they get so scared wondering which kind of illness they have, and this is where counseling is needed.* (Female vendor, high RDT use)

As explained in this quote, many vendors have never received formal training on counseling, and there is a specific need for counseling after negative RDTs. Some vendors also requested information about the diagnosis and treatment of other febrile illnesses. Vendors and clients reported experiences when typhoid, ulcers, flu, diarrhea, and pregnancy were mistaken for malaria.

The most significant challenge raised with a training intervention was how to include vendors who don't have the necessary educational qualifications to operate a drug shop in Uganda or whose shops are not licensed. Some vendors may be hesitant to participate in a government sponsored program because they fear having their shop shut down or their supplies confiscated. While some officials indicated their willingness to invite all drug shops, regardless of licensing status, one said that the district health office would only work with licensed shops. Other challenges mentioned were the large scale and costs, and the need for ongoing trainings to reinforce concepts and account for shop turnover.

A program to provide free or subsidized RDTs to drug shops was very popular with drug shop vendors and VHTs because it would make it easier for drug shops, which sometimes lack capital, to stock RDTs. Most strongly believed the proportion of clients who get RDTs at drug shops would increase once cost was no longer a barrier and vendors could provide better treatment recommendations as a result. Three vendors were concerned about losing profits if RDTs were free, but they accepted subsidized RDTs as a reasonable alternative that would still allow them to make money from malaria diagnosis.

*This program will help us to treat our clients when we know what we are treating, because each drug shop shall be testing for malaria before administering antimalarials. The program will also help to reduce on the number of patients that do self-treatment at home because the clients won't be fearing the cost of buying RDTs.* (Female vendor, high RDT use)

While vendors, like the one quoted here, had positive perceptions, health officials expressed more mixed views and recognized several potential challenges. This intervention would require significant funding and therefore external donors or partners to support the

government, which also raises concerns about sustainability. Additionally, there is a need for supervision and accountability, especially to ensure vendors are not selling the RDTs for their own profit, which a few vendors admitted might be tempting.

Increased enforcement of drug shop regulations was another intervention suggested only by health officials. Some believed that poor malaria case management in the private sector was largely a result of unlicensed drug shops being run by unqualified vendors, and they felt practices would improve if only licensed shops were operating. While Uganda has drug shop regulations, they are not consistently enforced. Health officials described challenges identifying shops in hard-to-reach areas due to the terrain, corruption, lack of political will, and inadequate funding. However, increased regulation could limit access to essential health services for community members that depend on drug shops. Even health officials with misgivings about vendor qualifications and practices recognized the potential for health centers to become overwhelmed if communities were unable to receive malaria testing and treatment at drug shops.

As an alternative to more punitive enforcement measures, some vendors, health officials, and VHTs suggested integrating drug shops into the public healthcare system.

> *In government facilities. . .our [malaria] testing rate is always above 90%. . .This is now what we actually need to work on, to see that these private ones also work according to how the government facilities are working. And I think if we can find ways of bringing these people on board, I think we can achieve this.* (District-level health official)

> *We as drug shop vendors, we are like people who operate illegally, because the government is always hunting for us. Therefore, the question is, does the government know how we are into existence and how we play a big role in the community? Why don't they give us peace to operate instead of harassing us? And in case we are not doing the right things, they need to come to us and guide us on what to do than harassing us always.* (Female vendor, medium RDT use)

These quotes highlight how improved public-private partnerships could benefit health officials and drug shop vendors. While some vendors were fearful or suspicious of working with the government, even suggesting invitations to participate in programs might be a "trap," many vendors expressed a desire for positive connections with the public sector. Specific suggestions included a formalized referral program for clients who test negative at drug shops to receive further testing and treatment at health centers, trainings that include providers from both the public and private sectors, or integrated reporting systems.

## Discussion

This study provides insights into the reasons for poor malaria case management practices at drug shops in Bugoye, Uganda, specifically the low use of RDTs and nonadherence to negative RDT results. We found that RDTs were valued and familiar to drug shop clients, in contrast to earlier qualitative research in Uganda, when very few individuals knew about RDTs [30]. Clients believed that RDTs are necessary to identify whether the cause of symptoms is malaria, and they liked that RDTs provide quick results that help you take the right medication and recover. However, clients often came to the drug shop with only enough money to purchase medications, and with limited resources, they prioritized treatment over testing. These findings are consistent with previous studies in sub-Saharan Africa that found client willingness to purchase an RDT was influenced by the cost of the test [29, 30, 41, 42]. Drug shop vendors described an increased demand for RDTs when community members have money available from coffee harvests, typically in March/April and September/October, which overlaps with

the periods of highest malaria transmission. This provides further support that clients do inherently value malaria testing, but financial barriers must be addressed to increase their use. While previous research documented a fluctuation in demand for private-sector antimalarials that corresponded with changes in household income due to the payment of school fees or profits from harvests [43], our study describes variation in RDT use at drug shops in Uganda according to the agricultural seasons.

Antimalarial purchases after testing negative for malaria stemmed from low confidence in the validity of negative RDT results, and clients' fear about what to do if they didn't have malaria, the treatable, familiar condition they were expecting to be diagnosed with. Similar to previous research in Uganda, mistrust in RDTs was largely due to expectations of false negative results among patients with low parasite/antigen loads (due to early detection of cases), previous antimalarial use, or malaria not caused by *P. falciparum* [44, 45]. Despite the frequency of previous experiences in our study where clients or their family members had an illness that was initially misdiagnosed as malaria, clients didn't see the disadvantages of taking antimalarials after a negative RDT. In these situations, recovery with antimalarials was seen as evidence that an individual had malaria the RDT failed to detect, reinforcing continued overtreatment.

We identified several promising intervention pathways that could improve RDT use and adherence to results. Primary among them is formal recognition of the essential role that drug shops play in addressing health challenges in their communities. Drug shops have emerged as the result of inequitable access to healthcare. In the presence of a myriad of financial, geographic, and logistical barriers to seeking treatment from public facilities, many people turn to the private sector. On weekends and in the evenings, drug shops are the only treatment option for many people. Efforts to address healthcare gaps and improve equity must switch the current framing of drug shop vendors as solely businesspeople to recognize their legitimacy as health providers that are trusted by the community. Interventions should acknowledge the important role that drug shops play in the distribution of antimalarials, and consider vendors 'partners in health' rather than targets of strict enforcement by the National Drug Authority [46].

Drug shop clients expressed high levels of trust in their vendors. While this confidence may be misplaced in some cases, it is a social resource that could be leveraged if drug shops are integrated into Uganda's healthcare system. The current relationship between the government and drug shop vendors in Bugoye is tense. Unlicensed drug shops often close their doors and hide when they hear the National Drug Authority is making inspections in the area, creating a situation where officials have little awareness of drug shop practices. Elsewhere in Uganda, government inspections of drug shops have been described as "unpredictable, authoritarian, and characterized by illicit payments [47]." If drug shops are incorporated into the health system as partners, there are greater opportunities for oversight and training. Additionally, lessons from the public sector could be used to increase RDT use at drug shops, and vendors would be appreciative of relationships more focused on improving care than cracking down on unlicensed shops.

Drug shop vendors expressed a desire to receive training to improve the health services they provide for the community, and trainings were an acceptable option to health officials and VHTs. While previous interventions in Uganda trained drug shop vendors on how to use and interpret RDTs [21–25], vendors in our study reported high levels of comfort using RDTs. Therefore, we recommend trainings focus on more nuanced aspects of RDT use, such as counseling. Most of the clients in our study who bought an RDT at the drug shop agreed to be tested based on the vendor's recommendation. Therefore, training on how to effectively describe RDTs to clients and encourage their use could influence client purchasing behavior and improve RDT uptake at drug shops. Vendors may also benefit from training on how to

appropriately counsel confused or anxious clients who test negative for malaria. Improved counseling at the drug shop that alleviates client concerns and ensures they understand their negative RDT result and the possible next steps for diagnosing and treating other conditions, may mean clients are less likely to resort to antimalarial use. Training could also provide vendors with information about illnesses confused with malaria that could be easily diagnosed with other RDTs at the drug shop, such as typhoid.

A program to provide free or subsidized RDTs to drug shops would address financial barriers to RDT use and was acceptable to vendors, VHTs, and officials. While this strategy has proven effective in increasing demand for RDTs in other settings in sub-Saharan Africa [48, 49], it would need to be combined with additional efforts to ensure adherence to RDT results. Additionally, modeling suggests subsidized RDTs in the private sector may be most effective in settings of high malaria transmission, where concerns about patients with malaria not receiving antimalarials (i.e., undertreatment) are more pressing than concerns about patients without malaria receiving unnecessary malaria treatment (i.e., overtreatment) [50]. RDT subsidies may also be most effective if they are financially advantageous to drug shop vendors [51]. Vendors in our study were amenable to subsidized RDTs if they still profited from the sale of the test and subsequent medication purchases. Still, the need for vendors to balance profit and best medical practices remains an unaddressed challenge with all suggested interventions.

While VHTs, vendors, and health officials often recommended health education and community sensitization as a strategy to improve RDT uptake and adherence at drug shops, interviews with drug shop clients suggest that further community sensitization may not be necessary. Clients were familiar with RDTs and recognized their importance, regardless of whether they had purchased an RDT at the drug shop or not. Therefore, education campaigns that emphasize the benefits of RDTs or remind community members that not all fevers are malaria, may do little to increase RDT uptake at drug shops. Given the frequency that RDT costs were cited as a primary and significant barrier, one promising avenue for community education is to emphasize the financial advantages of malaria testing. Messages could explain that RDTs may save time and money–clients will not waste money on ineffective or unnecessary medications, and they may be spared costly higher-level care if their disease is correctly identified and treated in its early stages, before symptoms become severe. To improve adherence to RDT results at drug shops, reassuring messages from trusted community leaders that attempt to alleviate fears associated with negative test results, and remind community members of their options for further testing and treatment, could be a nice complement to improved vendor training and counseling on this topic.

## Strengths and limitations

The strengths of this study include a strong theoretical foundation and the engagement of numerous diverse stakeholders. Purposive sampling was used to select interview participants and ensure voices reflected a range of experiences. Looking ahead to future interventions, multiple perspectives are needed to ensure recommendations account for diverse preferences and concerns. The study was limited by a small sample size, although consistency in key themes and findings across participant categories increases our confidence in results. Community perceptions of RDTs are limited by the exclusion of individuals who received treatment at health centers rather than drug shops.

## Conclusion

This study provides in-depth information on the root causes of overtreatment for malaria at drug shops in Bugoye, western Uganda, by exploring reasons for the low use of RDTs and

nonadherence to negative RDT results. While RDTs are valued by drug shop clients, with limited resources, clients prioritized treatment over testing. Mistrust in negative RDT results and fear about treatment options for conditions other than malaria led to nonadherence to RDT results. Public-private partnerships that recognize the essential role of drug shops, better incorporate drug shops into the healthcare system, and leverage the high levels of community trust in drug shop vendors are recommended. Interventions that address financial barriers to RDT use, emphasize the financial benefits of malaria testing, and train vendors on effective vendor-client counseling could increase RDT uptake and improve adherence to negative RDT results.

## Supporting information

**S1 Appendix. In-depth interview guides.**
(DOCX)

**S2 Appendix. Summary of drug shop practices in Bugoye.**
(DOCX)

**S3 Appendix. Interviewer summary and debrief form.**
(DOCX)

**S4 Appendix. Codebook.**
(DOCX)

## Author Contributions

**Conceptualization:** Victoria Shelus, Edgar M. Mulogo, Clare Barrington, Ross M. Boyce.

**Formal analysis:** Victoria Shelus.

**Funding acquisition:** Victoria Shelus, Ross M. Boyce.

**Investigation:** Victoria Shelus, Nobert Mumbere, Amos Masereka, Bonita Masika, Joackim Kiitha, Grace Nyangoma.

**Methodology:** Victoria Shelus.

**Project administration:** Nobert Mumbere.

**Supervision:** Victoria Shelus, Nobert Mumbere, Emmanuel Baguma, Rabbison Muhindo.

**Writing – original draft:** Victoria Shelus.

**Writing – review & editing:** Victoria Shelus, Nobert Mumbere, Amos Masereka, Bonita Masika, Joackim Kiitha, Grace Nyangoma, Edgar M. Mulogo, Clare Barrington, Emmanuel Baguma, Rabbison Muhindo, James E. Herrington, Jr., Michael Emch, Suzanne Maman, Ross M. Boyce.

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
