## [Decision Letter · Decision Letter 0]

8 Aug 2022

PGPH-D-22-01018

“Testing for malaria does not cure any pain” A qualitative study exploring low use of malaria rapid diagnostic tests at drug shops in rural Uganda

Dear Dr. Shelus,

Thank you for submitting your manuscript to PLOS Global Public Health. After careful consideration, we feel that it has merit but does not fully meet PLOS Global Public Health’s publication criteria as it currently stands. Therefore, we invite you to submit a revised version of the manuscript that addresses the points raised during the review process.

We look forward to receiving your revised manuscript.

Kind regards,

Abhinav Sinha, M.D.

Academic Editor

Journal Requirements:

1. We have noticed that you have uploaded Supporting Information files, but you have not included a list of legends. Please add a full list of legends for your Supporting Information files after the references list. 

2. In the online submission form, you indicated that "The data presented in this study are available on request from the corresponding author. Qualitative transcripts are not publicly available due to the consent provided by participants on the use of confidential data.". All PLOS journals now require all data underlying the findings described in their manuscript to be freely available to other researchers, either 1. In a public repository, 2. Within the manuscript itself, or 3. Uploaded as supplementary information.

Additional Editor Comments (if provided):

Reviewers' comments:

Reviewer's Responses to Questions

**Comments to the Author**

1. Does this manuscript meet PLOS Global Public Health’s publication criteria? Is the manuscript technically sound, and do the data support the conclusions? The manuscript must describe methodologically and ethically rigorous research with conclusions that are appropriately drawn based on the data presented.

Reviewer #1: Yes

Reviewer #2: Yes

2. Has the statistical analysis been performed appropriately and rigorously?

Reviewer #1: Yes

Reviewer #2: No

3. Have the authors made all data underlying the findings in their manuscript fully available (please refer to the Data Availability Statement at the start of the manuscript PDF file)?

Reviewer #1: Yes

Reviewer #2: Yes

4. Is the manuscript presented in an intelligible fashion and written in standard English?

Reviewer #1: Yes

Reviewer #2: Yes

5. Review Comments to the Author

Reviewer #1: Reviewer’s Comments

1. Grammatical errors throughout the manuscript, both in the abstract and the main text. Needs revision. Refer to lines- L36 (Suggestion: Replace ‘fevers presented’ with ‘fever cases presenting), L195-196 (consider revising), L206 (check the tenses), L544 – 545 (incomplete sentence)

2. Methodology: The study aims to capture the community perceptions about RDTs and to identify the barriers to use of RDTs. Hence, in-depth interviews should have been conducted with those community members as well who did not visit the drug shops. Not including these members of the community may induce bias in the study as their characteristics and reasons thereof for not using RDTs may be systematically different from clients coming to the drug shops for RDTs.

3. Sample & Recruitment: L139 mentions ‘recent clients’; was any cut off time period taken into consideration or any operational definition developed for the same?

4. Table 1: L164 – L167: While calculating the cut points for RDT sales, what was the denominator used in the previous studies? Does the word ‘clients’ include only the fever cases visiting the drug shops or all clients visiting the drug shops? This needs to be specified otherwise it may give erroneous information on the uptake of RTDs.

5. Study tools: L169 mentions that ‘discussion guides’ were used for the study; however, discussion guides are used for FGDs. More appropriate tool for the current study would have been interview schedule / interview guide as the method for data collection used in the study has been mentioned as in-depth interviews. Consider replacing the word ‘discussion guides’ with ‘interview schedule / interview guides’.

6. Strengths and weaknesses of the study may be cited in the manuscript.

7. References poorly compiled and not in accordance with either the Vancouver / ICMJE style. Page numbers missing in several of the references cited. Suggestion: Follow one particular reference style, either Vancouver or ICMJE style as per the journal’s requirement.

Reviewer: Dr Madhu Kumari Upadhyay

Professor

Department of Community Medicine

University College of Medical Sciences & GTB Hospital

New Delhi, India

Ph: +91-9871423042

Email: mu3071@gmail.com

Reviewer #2: This manuscript is of potential interest to readers as it expands the population's perspective on RDT and its adherence to malaria case management. In addition, the topic and findings are interesting and able to seek the attention of stakeholders to reframe the RDT use policies in the population, especially in the private sector.

However, the one limitation of this work is the small sample size while clustering the data. More in-depth research is needed based on the study observation. So, I would like to recommend accepting after revision.

it seems the study is part of a project, and details are missing. Please name the details and related study you mentioned on page number, line number 126. Please specify, that there is no overlapping of the reported data.

However, the authors need to consider the following important issues.

Introduction:

• Page number 3, Line number 62, please mention the exact incidence number reported in 2019-20. crosscheck with the WHO data.

• Page number 3, Line number 68, site of incidence Is not clear in the statement; please mention specific sites like Uganda.

• Some of the references are not directly related to the cited details and make the lengthy list Please refer to page number 3, line number 72. (5,7, 9-14), if the reference is not cited in other study details, omit them.

• Line Number 82. Authors have highlighted that "RDTs quickly diagnose malaria in settings where high-quality laboratory services are not routinely available and have significantly reduced antimalarial over-prescriptions" later statement needs to be checked as its not conclusive as appeared in the cited studies.

Method:

• The study did not mention participants' recruitment details and study designs. Please add some details about it.

• In Buyoge how many communities are living, and please give their brief details and their name, if possible, specifically the remotely placed communities.

• Line number 121, Please explain "select villages" as VHTs are supposed to work in all villages.

• Line number 125, Please specify the related study site details.

• In the sample size calculation, details of the total number of VHTs and targeted health officials in the communities are not mentioned.

• Page number 5, Line number 139, The term "recent clients is not clear; please elaborate.

• Please elaborate on how the sample size was calculated; it seems that the clustering of the participants was planned initially and then recruited participants. In clustering, the geographic diversity data is missing in drug shop clients and vendors. Please mention it.

• In page number 6, line number 147-148. If 96% of vendors are unlicensed, how do they maintain their clients' data records? Please mention how they maintain the client data.

• Page Number 7, table 1, citation of the previous study is missing on whose data, RDT sales cutoff developed.

• The tabulation form of the explored theme of all participant's categories can reflect a better understanding of the individual participant's category interview structuring.

• Please mention the interview guide details their level of expertise and if they were trained for the interview before the data collection. Please also mention

• A table with socio-demographic details (age, sex, education, residency status, family income) and health and community officials' categories of the participants can provide additional insights.

Ethical consideration:

The study mentioned monetary incentives to each health and community official. Please explain the reason and category of incentive. It was provided against the wage loss, travel, or any other allowance.

Result:

• Findings can also be mentioned in major subtheme, apart from identified themes, though it's mentioned in supplementary data.

• Page Number 9, line 211, you mentioned "initial point of care" please mention if its febrile status or preventive diagnosis. The study excludes febrile patients and severely affected patients with limited or no movement due to the disease.

• page number 10, line number 247, In the themes of perception of rapid diagnosis test VHTs members perception can be added in one or two statements to make it wider perspective.'

• Page number 10, line 263-264, mention the % along with the number for better clarity.

• Page number 11, line number 290, In Theme use of RDT you mentioned "Among the twelve clients who purchased RDTs", the number should be 13, as mentioned in table 1.

• Page number 13, line number 323, you highlighted the importance of income seasonality and malaria testing preference; please discuss in the discussion part whether the income and high transmission season overlap in the community.

• Page number 13, line number 346, information about antipyretic medication can be discussed if explored whether clients are considering the antimalarial as antipyretic.

• Page number 16, line number 414. In the Theme intervention recommendation author mentioned "how to include vendors who don't meet the necessary qualification" please discuss in the light of previous studies if any KAP survey was done. Please incorporate it in the discussion part.

In Conclusion:

On page number: 21 and line number 557, the author mentioned, "This study provides in-depth information on the root causes of overtreatment for malaria at drug shops in Bugoye". However, seems does not seem to match with the study's main findings; based on the two direct responses of the drug clients on their previous episode, it may not be drawn, rather should be "underuse of RDT". Please review the statement and reframe it.

References.

Some of the references are not in the prescribed format, please check it once again to make it uniform and systematic.

Please see:

Reference number: 7, 8. 10, 15, 35, 45

Some references are referring the supplementary data. Please check

19.

6. PLOS authors have the option to publish the peer review history of their article (what does this mean?). If published, this will include your full peer review and any attached files.

**Do you want your identity to be public for this peer review?** For information about this choice, including consent withdrawal, please see our Privacy Policy.

Reviewer #1: **Yes: **Dr Madhu Kumari Upadhyay

Reviewer #2: No

---

## [Decision Letter · Decision Letter 1]

7 Nov 2022

“Testing for malaria does not cure any pain” A qualitative study exploring low use of malaria rapid diagnostic tests at drug shops in rural Uganda

PGPH-D-22-01018R1

Dear Ms. Shelus,

We are pleased to inform you that your manuscript '“Testing for malaria does not cure any pain” A qualitative study exploring low use of malaria rapid diagnostic tests at drug shops in rural Uganda' has been provisionally accepted for publication in PLOS Global Public Health.

Best regards,

Abhinav Sinha, M.D.

Academic Editor

Although the authors have revised the MS as per the reviewers' comments, there still are some minor corrections to be made before publication:

1. The author can cite the open source for reference number: 35. It’s essential to access the related information mentioned there. As mentioned it was part of quantitative study, if its published ,better to share its citation.

2. The client registration form filled by vendors in the study time duration or wider duration, please mention the details and share the data collection form in the supplementary data, if possible.

3. L169: Insert word 'in' between experience and conducting

4. L573: Omit the word 'numerous'

5. L576: Omit the word 'a'

Reviewer Comments (if any, and for reference):

Reviewer's Responses to Questions

**Comments to the Author**

1. If the authors have adequately addressed your comments raised in a previous round of review and you feel that this manuscript is now acceptable for publication, you may indicate that here to bypass the “Comments to the Author” section, enter your conflict of interest statement in the “Confidential to Editor” section, and submit your "Accept" recommendation.

Reviewer #1: All comments have been addressed

Reviewer #2: All comments have been addressed

2. Does this manuscript meet PLOS Global Public Health’s publication criteria? Is the manuscript technically sound, and do the data support the conclusions? The manuscript must describe methodologically and ethically rigorous research with conclusions that are appropriately drawn based on the data presented.

Reviewer #1: Yes

Reviewer #2: Yes

3. Has the statistical analysis been performed appropriately and rigorously?

Reviewer #1: Yes

Reviewer #2: No

4. Have the authors made all data underlying the findings in their manuscript fully available (please refer to the Data Availability Statement at the start of the manuscript PDF file)?

Reviewer #1: Yes

Reviewer #2: Yes

5. Is the manuscript presented in an intelligible fashion and written in standard English?

Reviewer #1: Yes

Reviewer #2: Yes

6. Review Comments to the Author

Reviewer #1: Reviewer's comments have been addressed adequately, however few grammatical corrections needed as follows:

L169: Insert word 'in'between experience and conducting

L573: Omit the word 'numerous'

L576: Omit the word 'a'

Reviewer #2: I have reviewed the revised manuscript and found most of the queries were answered and incorporated changes, wherever needed. I would suggest that the author can cite the open source for reference number: 35. It’s essential to access the related information mentioned there. As mentioned it was part of quantitative study, if its published ,better to share its citation.

Additionally, the client registration form filled by vendors in the study time duration or wider duration, please mention the details and share the data collection form in the supplementary data, if possible.

7. PLOS authors have the option to publish the peer review history of their article (what does this mean?). If published, this will include your full peer review and any attached files.

**Do you want your identity to be public for this peer review?** For information about this choice, including consent withdrawal, please see our Privacy Policy.

Reviewer #1: **Yes: **Dr Madhu Kumari Upadhyay

Reviewer #2: **Yes: **Dr Piyoosh Kumar Singh
